# CEPS: An Open Access MATLAB Graphical User Interface (GUI) for the Analysis of Complexity and Entropy in Physiological Signals

**DOI:** 10.3390/e23030321

**Published:** 2021-03-08

**Authors:** David Mayor, Deepak Panday, Hari Kala Kandel, Tony Steffert, Duncan Banks

**Affiliations:** 1School of Health and Social Work, University of Hertfordshire, Hatfield AL10 9AB, UK; 2School of Engineering and Computer Science, University of Hertfordshire, Hatfield AL10 9AB, UK; aranp2010@googlemail.com; 3Department of Computing, Goldsmiths College, University of London, New Cross, London SE14 6NW, UK; harikalahkk@gmail.com; 4MindSpire, Napier House, 14-16 Mount Ephraim Rd, Tunbridge Wells TN1 1EE, UK; tony@qeeg.co.uk; 5School of Life, Health and Chemical Sciences, Walton Hall, The Open University, Milton Keynes MK7 6AA, UK; duncan.banks@open.ac.uk

**Keywords:** complexity, entropy, software, paced breathing, heart rate variability, HRV

## Abstract

Background: We developed CEPS as an open access MATLAB^®^ GUI (graphical user interface) for the analysis of *Complexity and Entropy in Physiological Signals* (CEPS), and demonstrate its use with an example data set that shows the effects of paced breathing (PB) on variability of heart, pulse and respiration rates. CEPS is also sufficiently adaptable to be used for other time series physiological data such as EEG (electroencephalography), postural sway or temperature measurements. Methods: Data were collected from a convenience sample of nine healthy adults in a pilot for a larger study investigating the effects on vagal tone of breathing paced at various different rates, part of a development programme for a home training stress reduction system. Results: The current version of CEPS focuses on those complexity and entropy measures that appear most frequently in the literature, together with some recently introduced entropy measures which may have advantages over those that are more established. Ten methods of estimating data complexity are currently included, and some 28 entropy measures. The GUI also includes a section for data pre-processing and standard ancillary methods to enable parameter estimation of embedding dimension *m* and time delay *τ* (‘tau’) where required. The software is freely available under version 3 of the GNU Lesser General Public License (LGPLv3) for non-commercial users. CEPS can be downloaded from Bitbucket. In our illustration on PB, most complexity and entropy measures decreased significantly in response to breathing at 7 breaths per minute, differentiating more clearly than conventional linear, time- and frequency-domain measures between breathing states. In contrast, Higuchi fractal dimension increased during paced breathing. Conclusions: We have developed CEPS software as a physiological data visualiser able to integrate state of the art techniques. The interface is designed for clinical research and has a structure designed for integrating new tools. The aim is to strengthen collaboration between clinicians and the biomedical community, as demonstrated here by using CEPS to analyse various physiological responses to paced breathing.

## 1. Introduction

Every researcher has a dream. Ours has been to find or create an easy-to-use, versatile and reasonably comprehensive toolbox specifically for the analysis of complexity and entropy in physiological signals (CEPS). Since 2011, in our own research on physiological signals and measures, including electroencephalography (EEG), heart rate variability (HRV), temperature, postural sway and respiration, we (D.M. and T.S.) have felt hampered as clinicians by the lack of such a non-specialist software package that would enable us to explore, compare and combine changes in a variety of complexity and entropy measures in response to interventions such as electroacupuncture or biofeedback. Here, we describe the toolbox that we have developed to suit our own requirements, and hopefully those of other researchers as well—although we are well aware that different research groups will have different priorities and interests. We also illustrate its use in analysing the effects of paced breathing (PB) on variability of Electrocardiogram (ECG), photoplethysmography (PPG) and respiration (RSP) data in a small pilot study.

We began by conducting an online review of existing physiological time series data analysis packages. Although we found that quite a number are already available, many are now dated, until recently the majority have required knowledge and programming skills that many clinicians and healthcare workers do not have, and most offer only limited analysis of complexity and entropies, our own primary focus. Thus, although 34,647 studies were found in PubMed with the search term ‘entropy’, less than 1.5% of these (508) are clinical studies; while of 1,734,525 returns for ‘complexity’, less than 1.4% (22,768) are clinical studies, and for ‘chaos’ less than 0.5% (only 65 of 13,493 returns) are clinical studies (figures retrieved 27 July 2020). This has provided the motivation for a number of available software tools that use a graphical user interface (GUI), more accessible to those without a computer science background [1,2,3,4,5,6,7,8,9,10]. Some of these have a particular focus on data acquisition [6] pre-processing [2] or filtering of data [4,11]. The majority are dedicated to analysis of HRV [7,10,12,13,14,15], with a few designed specifically for investigation of the EEG [16,17,18,19,20]. Many HRV analysis tools—such as Kubios HRV [13], HRVAS [21], HRVFrame [14], gHRV [7], HRVAnalysis [15], RHRV [22,23], RR-APET [10] and PyBioS [24]—include a selection of complexity and entropy methods, although some older ones offer only a single such measure [13,25,26], and others such as PyHRV are not in GUI form [27]. We found only one EEG-specific software whose main focus is on complexity or entropy measures [5], although a complexity toolkit for resting state functional magnetic resonance imaging (fMRI) was located, also containing a handful of entropy measures [28]. In addition, there are also some older more general time series analysis toolkits with a particular emphasis on nonlinear dynamics, such as the Chaos Data Analyzer [29], TISEAN [30] and MATS [31], with, most recently, EZ Entropy [9]. Obviously, fashions change and these toolboxes were all designed in keeping with the original interests and objectives of their developers [32]. None of them completely met our own requirements for an easy-to-use package that would enable preliminary parameter testing, simultaneous analysis of many different multiscale measures, comparison and classification of results. Those that came closest were EEGFrame (22 nonlinear measures, including eight entropies) and HRVFrame (21 nonlinear measures, including six entropies), with their successor MULTISAB (unfortunately not open access) [33], HRVAnalysis (around 12 nonlinear measures, including six entropies), PyBioS (an object-oriented tool for modelling and simulation of complex biological systems, with some 10 nonlinear measures, including seven entropies) and EZ Entropy (six entropies, but no other nonlinear measures). The latter two were explicitly designed to be used with non-time series data in addition to physiological signals such as HRV. Only Kardia [26], Kubios HRV [13], RHRV [22,23], RR-APET [10] and PyBioS [24] provided more than a cursory introduction to the complexity/entropy measures offered. The packages reviewed are listed in Appendix A, showing which measures are implemented in each, as well as citation counts from Google Scholar and SCOPUS. The majority of the packages are open-source GUIs, although fewer offer batch processing. A number provide useful pre-processing and data segmentation possibilities, but only a handful include multiscale analysis or classification modules.

Beyond those packages mentioned above, our literature search found only one additional tool which included some of the complexity and entropy measures that were of interest to us, published in 2020 by Mendonça et al. [34], but this was for a very limited application, was not GUI-based, and was far too technical for a non-specialist user. Yet, over the past three decades, more and more scientific studies have concluded that traditional, linear methods of analysis are insufficient for complete analysis of the underlying patterns that occur in complex physiological signals [9,35,36]. This has now become widely accepted and is rarely questioned. We also conducted a brief review of published studies to see if this is really the case and not just a matter of opinion (Appendix A). Our bibliometric review was encouraging. Furthermore, while conducting our review we found many indications that different complexity and entropy measures provide different information and so could be considered complementary [37], suggesting that multiple measures should be used [38], not just a single one, particularly when analysing concurrent changes in several physiological signals [39].

We therefore felt justified in developing our own free, open-source GUI for a good variety of complexity and entropy measures (in particular, multiscale measures), of which there are now so many. As the name CEPS suggests, these measures are rather like the fungi Cèpes (Boletus edulis)—deliciously fascinating, appearing mysteriously and apparently out of nowhere. You just have to turn your back on the internet for a moment, and there’s another one you’ve never seen before. The challenge is to come to know which are worth exploring further, and which are indigestible. We hope that CEPS will facilitate this process but should emphasise that researchers will still have to make their own, careful decisions, and not use the GUI as a substitute for critical thinking. To help make this paper more digestible, a list of the many abbreviations used is included in the table at the end of the article.

### Objectives

Our objectives of the paper and software are as follows:To explore the literature on complexity and entropy measures for those suited to a MATLAB^®^ GUI for users who may not be familiar with them and also not expert in computer programming methods.To investigate the ‘family trees’ of different complexity and entropy measures in order to better select existing codes for inclusion in CEPS.To create a GUI that would allow univariate analysis of single and multichannel (time) series data, specifically for—but not limited to—the data types resulting from our own research. CEPS should thus include some simple analytical methods and basic normality tests, as well as methods to calculate complexity and entropy measures, multiscale measures in particular.For CEPS to include some basic pre-processing steps and some ancillary methods for estimating embedding dimension and time delay parameters. Batch processing (import and export) should be possible, and different import and export formats catered for.As a central feature, for CEPS to include a ‘Test and Plot’ facility for experimentation with measure parameters prior to processing and exporting results. As a priority, to test CEPS with reference to results obtained using other packages, although this is not always a simple process, given the variety of parameter settings possible [7]. Where this is not feasible, to verify results in consultation with the originators of the different complexity and entropy measures implemented.Eventually, for CEPS to include a final ‘Classification’ section where results using different measures and methods can be compared, and a ‘plug-in’ facility to allow other researchers to add measures not already included in the list available.To include a ‘Primer’ with CEPS containing enough background information and references to enable those unfamiliar with the concepts of complexity and entropy to use the GUI and process their own data without too much of a ‘garbage in—garbage out’ result.To illustrate how CEPS may be applied in practice by using the GUI for analysing the effects of paced breathing (PB) on variability of ECG, PPG and RSP data in a small pilot study. In particular, to compare the relative performance of conventional linear (time- and frequency-domain) indices and the nonlinear measures provided in CEPS, as well as between the different complexity and entropy measures.

## 2. Materials and Methods

### 2.1. Literature Review

A literature search was conducted in PubMed for relevant software packages. Search terms were drawn from keywords found in the titles and abstracts of some of the packages mentioned in the Introduction, which had already been located through prior knowledge, searching references or personal recommendations. These terms were concatenated into a preliminary search string:
(tool[Title] OR toolbox[Title] OR “graphic* user interface”[Title] OR Software[Title] OR program*[Title]) AND (signal*[Title] OR series[Title]). This resulted in 1190 papers for the period 2000–2020.

Adding AND (ECG OR HRV OR EEG OR temperature OR respiration) reduced this number to 74, of which 17 were not known to the searcher (D.M.) but were judged to be of interest from their Abstracts. Most of these turned out to be for very specific applications, beyond the aspirations of CEPS (see Introduction for a brief summary of results).

A further search of PubMed was made, on the basis of prior knowledge and measures used in existing software packages, in order to locate studies on complexity and entropy measures likely to be useful in CEPS. The generally accepted names of the complexity and entropy measures were used as search terms, as shown in Table 1 and Table 2.

Many of the studies located used multiple measures, some not mentioned in their titles or abstracts, enabling further searches for measures not previously encountered. When it became clear that further searching was unlikely to turn up any newer measures, the numbers of studies on each measure were counted, and those measures used infrequently were not considered further. Results are included below in Section 3.1.

A review was also undertaken from published studies of the data requirements for all these measures (discrete and/or continuous, short and/or long samples), the measures’ robustness to noise, whether they may be (or have been) used with bandpass (or other) filtered data, if they are affected by sampling rate or down-sampling, and whether data needs to be stationary or linear if the measures are to be used appropriately. The results of this review are summarised in Section 3.1.2 below. Information on parameter settings was also gathered, and, for some but not all measures, their expected values where possible. These are presented in the reference document *Primer on Complexity and Entropy* that is accessible via the HELP section in the CEPS GUI.

After preparing a shortlist of measures to be considered for inclusion in CEPS, further PubMed searches were undertaken, using as search terms the names of the researchers who had contributed most papers on each complexity and entropy measure. Results were then collated, and researchers ranked both by the number of their published studies and by how many different measures they had investigated. MATLAB codes used by the highest-ranking researchers or others from their university departments were then sought, for use in CEPS. In recent years, some researchers have generously published the code used in their work, or at least a pseudocode that can then be translated and used by others. When neither was available, code was requested from its originators. The codes used in CEPS and their sources are listed in a Table in Section 3.1.2, with further information provided in an Appendix to the CEPS *Manual*.

In addition, a brief literature review was undertaken using PubMed to locate studies on PB and its effects on entropy or other nonlinear measures in ECG or RSP data.

### 2.2. Creating CEPS

CEPS was developed in MATLAB, initially with GUIDE (‘Graphical User Interface Development Environment’), which was further enhanced by migrating the GUIDE application to App Designer. On the basis of our previous experience in analysing HRV and EEG data, an initial list of descriptive and simple linear measures was created first (17 items), together with some standard normality tests and time- and frequency-domain measures (15 items). Stationarity tests and a nonlinearity measure [40] were then implemented (six items). The main list of complexity and entropy measures was built using our literature reviews, with 10 and 28 items, respectively, together with three methods of symbolic dynamics. Standard ancillary methods for estimating embedding dimension and time delay were included (three items). The interface was designed in stages, and new measures were tested against results from other published software tools or, where this was not possible, by seeking verification from the methods’ originators. Methods of verification used are shown in a Table in Section 3.1.2. Testing and debugging are still in process, incorporating user feedback from a usability survey, following the example of gHRV [7] and RR-APET [10]. The survey form is accessible via the GUI.

### 2.3. Paced Breathing Data

Electrocardiogram (ECG), photoplethysmography (PPG) and respiration (RSP) data were collected from a sample of nine healthy adults in a pilot for a larger study investigating the effects on vagal tone of breathing paced at various different rates, part of a development programme for a home training stress reduction system managed by MindSpire and funded by Innovate UK (reference 63382). Participants were informed about the study and signed a consent form with the explicit agreement that their anonymised data would be stored in perpetuity on a publicly available repository for other researchers to use. Ethics approval procedure was followed, according to the principles of the Helsinki declaration.

In this pilot, two 300-s trials were recorded for each person. The first was a baseline, in which people were instructed to sit comfortably and breathe normally. In the second trial, a few minutes later, they were instructed to breathe in and out, guided by a laptop pacer display, with a coloured bar growing and shrinking onscreen at seven cycles per minute.

Single-lead ECG signals were obtained from one electrode applied on the volar surface of each forearm, with the ground electrode on the volar surface of the left forearm, PPG signals from the middle finger of the left hand, and respiration from a SleepSense abdominal respiration belt with a piezoelectric crystal effort sensor. Data were collected using a NeXus-10 amplifier with BioTrace+ software v 2015B (Mind Media, Herten, The Netherlands), exported at 256 Hz. The ECG signal was sampled at the same frequency, PPG at 128 Hz and RSP at 32 Hz. Data were exported from BioTrace+. The ECG and PPG signals were then processed separately using industry-standard HRV software, Kubios HRV Premium software (v3.1; Kuopio, Finland), with an automatic RR correction algorithm to deal with artefacts and a ‘smoothness priors’ method of trend removal [41]. For spectrum estimation, a piecewise cubic spline interpolation was used and the default resampling rate of 4 Hz. The graphed output from the Kubios HRV software for each of the resulting recordings was then examined carefully for any remaining unusual findings or artefacts.

In addition to using Kubios HRV Premium for heart and pulse rate analysis, as part of our suite of tools for physiological data analysis, one of the present authors (D.P.) had earlier developed a versatile MATLAB-based GUI for peak detection in raw ECG, Photoplethysmograph (PPG) and Respiration time series data. This was used here for peak detection in the RSP signal, again followed by examination of each data file and manual correction if appropriate. The resulting R-to-R beat interval (RRi) data from the ECG, PPG pulse-to-pulse intervals (PPi) and RSP inter-breath intervals (BBi) were then used as input for CEPS. All measures in CEPS that were suitable for short datasets were selected for further analysis, using default parameters. The usual time- and frequency-domain HRV measures, as well as three derived from Poincaré plot analysis (SD1, SD2 and their ratio) were also computed for the three data streams using Kubios HRV Premium. Standard frequency ranges were set: ‘Very low frequency’ (VLF), 0–0.04 Hz, ‘Low frequency’ (LF), 0.04–0.15 Hz, and ‘High frequency’ (HF), 0.15–0.4 Hz.

Analysis was conducted in MATLAB R2019a, Excel Version 2011, SPSS Version 26 and RStudio Version 1.3.1093. Bootstrapped paired t-tests were used to check for significant differences in measures between the ‘baseline’ (unpaced) and paced breathing conditions. Case-resampling bootstrap was used, with 1000 samples, resulting in percentile-based *p*-values.

Complexity and entropy measures are essential for quantifying nonlinear physiological processes, and in particular the changes in their dynamics that occur over time. To properly encompass the multifarious complexity of physiological signals also necessitates, we believe, considering a large number of those variables that are appropriate for analysis (see Table 3). In this analysis, we have used the Benjamini–Hochberg procedure to decrease the false discovery rate and so reduce the Type I errors (false positive results) that inevitably occur when conducting the resulting multiple comparisons.

A false discovery rate (FDR) of 0.12 was selected, to provide usable *p*-values for around 50 (just over a quarter) of our 198 measures (50, 45 and 48 measures, for ECG, PPG and RSP, respectively), with uncorrected *p*-values of 0.033 or less. When an FDR of 0.05 was selected, none of the 198 measures appeared to give positive results, which was patently not the case, whereas for FDR = 0.10 (often used in screening studies), between 21 (for RSP) and 38 (for ECG) of the measures did (35 for PPG). For FDR = 0.2, on the other hand, around 100 measures resulted in *p*-values less than a Benjamini–Hochberg critical value of 0.05 for each data stream (109, 94 and 108, for ECG, PPG and RSP, respectively). The choice of FDR = 0.12 seemed a suitable compromise.

Cohen’s *d* was used as a measure of Effect size [42]. Effect size (Cohen’s *d*) and rank or corrected *p*-values are closely related. Here, we have chosen to report effect size, followed in square brackets by the range of *p*-values that remain significant after applying the Benjamini–Hochberg procedure.

## 3. Results

### 3.1. Literature Review

#### 3.1.1. Complexity and Entropy Measures of Potential Interest Located in PubMed

These measures are shown in Table 4 and Table 5, together with references for software packages known to include them and various other bibliometric details. 

For further review results, see Section 3.1.2 below, and the *Primer on Complexity and Entropy* that is accessible via the HELP section in the CEPS GUI.

#### 3.1.2. Researchers, Institutions, and Measures

The most prolific researchers on complexity or entropy—those with more than 25 papers on these topics indexed in PubMed—are listed in Table 6, showing the numbers of their papers indexed, the number of different complexity or entropy measures mentioned in the abstracts to those papers and listed in Table 4 and Table 5 above, the date their first relevant paper appeared in PubMed and the University or other institutions with which they were affiliated then and—if different—with which they were affiliated when their most recent papers were published. Note that the numbers of measures and papers shown will have increased now that the journal *Entropy* is indexed in PubMed. More detailed information can be found in the Appendix A.

The earliest publication dates for each researcher fall into four groups: 1991–1995 (8), 1996–2000 (7), 2001–2005 (8) and 2006–2011 (8). The four most established ‘specialist’ researchers (with papers on three measures or less located via PubMed) appear to be Ary Goldberger and Steven Pincus (1991), Li Peng (1992), and Johannes Veldhuis (1994), the four most established ‘generalists’ (with papers on 13 or more measures in PubMed) being Andreas Voss (1991), Alberto Porta and Maria Signorini (1994), and Jamie Sleigh (1995).

Most institutions occur only once in this Table, apart from Harvard (2) [Goldberger, Peng], Leiden (2) [Frölich, Roelfsema], Melbourne (2) [Karmakar, Palaniswami], National Taiwan (2) [Lo, Shieh], Oulu (2) [Huikuri, Mäkikallio] and Valladolid (4) [Abásolo, Escudero, Hornero, Poza]. More than half of the researchers listed (17 of 31) remained at the same institutions over the period.

Further information on the interconnections between some of these institutions and between the different complexity and entropy measures investigated at them can be found below, in Figure 1 and Figure 2 and Table 7.

Notes
1.For 1-dimensional data, H = 2–FD [89].2.In certain situations, DFA α is directly related to fractal dimension D, with D = 2–α/2 [90].3.For 1-dimensional data, H ≈ DFA α [91].4.D_2_ is related theoretically to the Lyapunov exponents [92].5.The LLE can be estimated from RQA [93].6.See [94]7.Pedro Bernaola-Galván, the main originator of the volatility method of assessing nonlinearity, has co-authored work on nonlinearity with Alberto Porta [95].8.Paolo Castiglioni, whose code for mFmDFA is used in CEPS, has co-authored at least six papers with Alberto Porta.9.Paolo Grigolini, the originator of Diffusion entropy, co-authored some papers on diffusion with Constantino Tsallis when both were at the University of North Texas [96].10/11.SE, CE and DiffEn were all introduced by Shannon in his famous 1948 paper [97].12.Corrected CE (cCE) and corrected ApEn (cApEn) were both introduced by Alberto Porta and his associates [72,98].13/14.ApEn [99], D_2_ and LLE can be estimated from RQA [100].15.Both AF and DFA are methods of assess the fractal exponent α, the former particularly for point process data [101].16.KSE can also be estimated from RQA [102].17.See [103].18.See [104].19.See [105].20.See [106].

Family trees are always incomplete, and this one is no exception. It is the result of literature review, but not a systematic one, so represents a personal selection by the authors. While in the final stages of preparing this paper for publication, we became aware of a similar—if more technical—analysis of the ‘entropy universe’ by the indefatigable Teresa Henriques and her colleagues in Porto [107]. Readers will find it instructive to compare our two mappings. CEPS does not include the family (or ‘galaxy’) of topological or graph entropies, for example [108], although we do plan to include some applications of maximum entropy.

#### 3.1.3. Paced Breathing

Paced breathing is used here as an example dataset to display the advantages of the CEPS data analysis tools when focusing on complexity and entropy measures. Our analysis investigates the effects of paced breathing on ECG, PPG and RSP variability.

For millennia, slow deep breathing at around 4 to 10 breaths per minute (0.07–0.16 Hz) has been thought to promote both physical and mental health benefits and improve wellbeing. Research over the past few decades has supported this belief and shown a number of beneficial physiological effects, for example improving ventilation efficiency and blood oxygenation (SpO_2_) in hypoxic patients [142], lowering blood pressure in the overtly healthy [143], reducing experimental pain perception [144], and lessening anxiety scores in school children [145].

By measuring the HRV during a stepwise-paced breathing procedure where participants are asked to breathe at rates ranging from 14 to 4 breaths per minute (bpm), an individual’s respiratory ‘resonant frequency’ where the HRV is greatest can be determined and is generally around 6 bpm (i.e., 0.01 Hz) for adults [144].

Slow breathing is known to affect both time- and frequency-domain measures of heart rate variability [146]. However, less is known about how it impacts entropy and complexity measures. In one study using 3-min recordings, DFA α2, ApEn, SampEn and MSE were reduced during slow paced breathing, while DFA α1 increased [147]. Others too have noted a reduction in MSE with slow paced breathing (at least in flight phobics, but not in healthy controls) [148]. Porta’s research group, on the other hand, found that heart period ‘multiscale complexity’ (but not MSE) decreased with metronome-paced breathing at 10 bpm as a result of regularisation of variability in both LF and HF HRV bands [149]. An associated research group noted that paced breathing at 12 bpm enhanced HRV nonlinearity, and that SampEn decreased significantly when moving from spontaneous breathing to breathing paced at 12 bpm (0.2 Hz), but was substantially unchanged between spontaneous breathing and breathing at 18 bpm (0.3 Hz), when cardiorespiratory synchronisation competes with sinus arrhythmia [150]. Perhaps the most informative of these studies is one in which breathing was paced at between 6 and 27 bpm. SampEn in both the ECG and RSP signals, as well as the ECG RRi itself, increased with respiratory frequency between 6 and around 15 bpm, but results were affected by both age and sex [151].

### 3.2. A Brief Description of CEPS

As mentioned, CEPS is an open access MATLAB™ GUI for analysing **C**omplexity and **E**ntropy in **P**hysiological **S**ignals. It includes data analysis tools that focus on those complexity and entropy measures that appear most frequently in those peer-reviewed publications that are indexed in PubMed, as well as others that are less well known.

#### 3.2.1. Installation

CEPS, together with a Manual and Primer on complexity and entropy, is publicly available at https://bitbucket.org/deepak_panday/pipeline/src/pipeline_V2/ (accessed on 11 March 2021) for free, non-commercial use, although MATLAB must be installed. The Manual and Primer are also accessible as *.pdf files via a drop-down list in the HELP section of the GUI, as are a feedback form and the present article.

#### 3.2.2. Loading Data

When the GUI first opens, the HELP section and a second drop-down list of six Application Modes appears. The six modes, which should be used in order, are: Load Data, Pre-Process Data, Test Parameters, Run Pipeline, Process Results and Classification (the latter is not yet implemented). To proceed, data should be ready for processing and stored in a single location. The ‘Load Data’ button is used to navigate to that location. Data should be univariate but can be in single or multiple column format, with *.txt, *.csv, *.mat or *.xlsx file extensions. The number of files in the chosen location with the desired extension will appear in the Total Data Files box. Once data are loaded using the ‘Load Data File/s’ button, the GUI is ready for use.

In the Data Type pane (at the top left of the GUI), Time Series or Non Time Series Data can be selected. For the former, a Sample Rate (sampling frequency) in cycles per second (Hz) will need to be selected, and if data is to be subdivided into non-overlapping epochs for analysis, an Epoch Length in seconds will also need to be specified. For Non Time Series Data, the Number of Data Points per Epoch should be entered.

#### 3.2.3. Pre-Processing Data

CEPS can process raw physiological data like EEG or temperature, but in order to process the periodic RSP BBi we used the Signal Processing MATLAB GUI mentioned above, results being opened directly in CEPS to compute the complexity and entropy Measures.

When the Pre-Process Data mode is selected, the list of eighty-two Measures below the Application Mode selection box is inactivated, but the central Panel on the right (II) will show two plots (Figure 3).

Various pre-processing methods are available, including outlier removal, filtering and adding different types of noise. In a forthcoming version of CEPS, it will also be possible to detrend or rescale data (using min-max normalisation) or to use Z-score normalisation (standardisation), as in EZ Entropy [9] and PyBioS [24]. In addition, as many entropy measures are more appropriate for discrete rather than continuous data (see Table 8), a selection of coarse-graining methods will be provided, as well as interpolation and resampling as a basis for frequency domain analysis. More details are provided in the Manual, and users need to ensure that data is appropriately pre-processed before moving on to testing parameters or running the pipeline.

#### 3.2.4. Testing Parameters

A serious problem with many complexity and entropy measures is how to select the parameters used to estimate them. Using fixed parameters may not work well all the time [38]. A ‘test and plot’ facility is therefore provided to assist in both parameter selection and multiscale analysis, for single or multiple files. In the Test Parameters mode, only the 46 measures in the list on the left of the GUI which require parameter settings can be selected using the check boxes provided, and only one at a time. Selecting a measure in the list will bring up *two* boxes for each parameter required in the central panel of the GUI (Run Parameter Test II). When the measure is selected, the name of its parent section in the Measures List replaces ‘Run Parameter Test II’ as the name of the central panel. A range of values for each parameter can then be tested, again for only *one* parameter at a time. Care should be taken to enter values that will give meaningful results (guidelines are provided in the Manual).

When a particular measure is selected for parameter testing, the user can choose whether to test some or all of the loaded datasets, some, or all of the epochs in those files, and which parameters to test. Parameter increments between the minimum and maximum values chosen can also be set. Whether integers or decimals are entered here will depend on the Parameter being tested. Results can be plotted by clicking on ‘Plot Result’ and, if required, displayed in a Data Table as well. Both plot and table are shown in the central area of the GUI, and both may be exported, so it is possible to double check the effects of selecting different parameters on the behaviour of the measure investigated (Figure 4). Computation Time is shown in the lower pane, ‘Run Parameter Test III’.

#### 3.2.5. Running the Pipeline

Both when testing parameters and running the pipeline, in the upper pane (‘Pipeline Execution I’) the user can select whether to process either a complete data series or some or all of the epochs into which it may be subdivided.

When running the pipeline, in contrast to when testing parameters, several measures can be selected in the Measures List at the same time, according to user requirements and research design, the main limitation being computer processing speed. However, now only one value can be assigned to each parameter for a particular measure, and this value will be used for all data files processed (hence the importance of testing parameters in order to make an informed decision about which values to use). As when testing parameters, options can be selected in the ‘Pipeline Execution II’ pane. A full list of the measures available is given in Table 7.

#### 3.2.6. Processing Results

In the ‘Process Results III’ pane (Figure 5), results can be saved in Excel and/or MATLAB format. Parameters used for the various measures will be saved in the same file in separate sheets, as will Computation time if this option is selected. For batch processing, the option is given to aggregate results by mean and standard deviation or median and interquartile range.

#### 3.2.7. Classifying Results

This Mode—planned, but not yet implemented—will be useful when batch processing. For binary groupings of individuals (normal vs. paced breathing, for example), the different measures can be assessed and compared for their sensitivity and selectivity in classifying the individuals in each group (Simple classification). When more than two categories or measures are considered, Compound classification is being explored, perhaps offering options for *k*-means prediction, AdaBoost and Linear discriminant analysis. Results for the different groupings can also be displayed in exportable plots.

#### 3.2.8. System Requirements, License, Troubleshooting and Sample Data

CEPS was developed initially using MATLAB^®^ 2019a. Existing tried and tested MATLAB functions were implemented where possible (sources indicated in Table 8). Currently, MATLAB must be installed before CEPS can be used, but a compiled version is foreseen as a future development. A Windows or Linux 64-bit or Mac operating system is required, and for fast plotting and calculation, 8 GB of RAM and a screen resolution of 1400 × 900 pixels are recommended. The complete download will take 800 MB of disk space. As for other MATLAB-based GUIs developed using App Designer [9], MATLAB^®^ 2018a or later should be used for best performance.

The GUI can be downloaded from Bitbucket. CEPS is open access and free to non-commercial users, under the terms of the GNU General Public License, version 3 or later. Open access MATLAB scripts come with permission to redistribute, with or without modification, and a standard warning that they cannot be assumed to be fit for a particular purpose. Further license details are available in the Manual.

Error messages may appear in the MATLAB Command Window or as pop-ups in the GUI itself. If you use CEPS and find a persistent problem, do feel free to report it to one of the authors (D.M. or D.P.), or on the BitBucket site.

Fifteen files of RRi and 19 files of unprocessed EEG data (300 s long, both in *.txt format) are packaged with CEPS, as well as a number of synthetic data samples. Users can of course also build their own libraries of sample data.

### 3.3. Paced Breathing Data—Some Basic Analysis

Of our nine participants, five were women. Ages ranged from 16 to 55 (mean 39, SD 16). Their 5-min RRi and PPi recordings were all 320 data points long (mean 319, SD 2.15), and the RSP data was around 50 breaths long (mean 48.4, SD 19.0). Of our 198 measures, most were normally distributed in both normal and paced breathing (more measures were not normally distributed during paced than during non-paced breathing).

Paired t-tests with FDR = 0.12 indicated the following numbers of significant increases and decreases in measures for the three different data types: 15 increases and 35 decreases for the ECG data, 16 increases and 29 decreases for the PPG data, with 11 increases and 37 decreases for the RSP data. The highest-ranking ECG and RSP measures (i.e., those with lowest effect sizes) were then removed from further analysis to ensure equal numbers of measures (45) were compared for the three data types. Two complexity, one entropy and two frequency-domain measures were removed for ECG and one complexity, one time-domain and one frequency-domain measure for RSP, leaving 13 increases and 32 decreases for ECG, 16 increases and 29 decreases for PPG, and 11 increases and 34 decreases for RSP.

#### 3.3.1. Measures That Increased during Paced Breathing

Of the 13 remaining ECG measures that increased during paced breathing, 6 were measures of complexity from CEPS, and 7 were frequency measures from Kubios HRV. Median effect size was 1.894 for the former, reducing to 1.724 if ACN at lag 10 was included [*p*-values ranging from 0.006 to 0.027]. For the frequency measures, median effect size was less, at 0.946 [*p* from 0.012 to 0.025].

Corresponding findings for the PPG data were similar: 7 were measures of entropy or complexity from CEPS, and 9 were frequency measures from Kubios HRV. Median effect size for the former was 1.454 [*p* from 0.004 to 0.025], and 1.013 [*p* from 0.012 to 0.031] for the frequency measures.

For the RSP data, in contrast, median effect size for the seven HFD measures showing significant increases (kmax = 5 and from 9 to 14) was 1.292 [*p* from 0.007 to 0.029], whereas for the three linear measures from CEPS that increased significantly, median effect size was greater, at 1.643 [*p* from 0.006 to 0.022]. Effect size for the single Kubios HRV time-domain measure that increased significantly (mean RRi) was 1.297 [*p* = 0.028], similar to those for HFD.

HFD at various lags increased consistently for all three data types (3 for ECG, 4 for PPG and 7 for RSP), with median effect sizes 1.894 [*p* from 0.006 to 0.013], 2.035 [*p* from 0.004 to 0.011] and 1.292 [*p* from 0.007 to 0.029], respectively. Cohen’s *d* was particularly high for RQA ENT (Shannon entropy of the line length distribution in the recurrence plot) in the ECG data, with a value of 2.214 [*p* = 0.003], but although *d* was also high in the PPG data (2.239, *p* = 0.002), this was not significant after applying the Benjamini–Hochberg procedure. RQA ENT in the RSP data decreased significantly between baseline and paced breathing, with effect size 0.814 [*p* = 0.029].

#### 3.3.2. Measures That Decreased during Paced Breathing

Many more measures decreased than increased with paced breathing for ECG and RSP, but not for PPG. As the measures were not all independent of each other, there is no readily available statistical test to assess the significance of differences between the numbers of those that increased and decreased. However, as an indication, the *p*-value associated with the Binomial test (a test of independent measures) would yield values of 0.007 for ECG, 0.001 for RSP, but only 0.072 for PPG.

For ECG, median effect size for the 25 complexity and entropy measures that decreased significantly was 1.991 [*p* from to 0.004 to 0.033], reducing to 1.903 if four ACN lags are included, whereas for the four frequency-domain measures showing significant differences it was less, at 1.035 [*p* = 0.008 or 0.029], and for the Kubios Stress Index [153], it was only 0.500 [*p* = 0.029].

For PPG, median effect size for the 23 complexity and entropy measures that decreased was 1.933 [*p* from 0.003 to 0.033] and for Hjorth complexity (a linear measure) it was 1.419 [*p* = 0.026], whereas for the 5 frequency-domain measures it was only 0.901 [*p* from 0.005 to 0.027].

For RSP, results were a little more complicated. Median effect size for the 24 complexity and entropy measures that decreased significantly was 1.483, decreasing to 1.475 if the one significant ACN lag is included [*p* from 0.008 to 0.030], whereas for the two frequency-domain measures that decreased significantly it was 0.801 [*p* = 0.007, 0.010], and for the three Kubios time-domain measures that decreased significantly it was 1.546 [*p* from 0.006 to 0.024]. For the linear measures in CEPS and for Skewness, it was 1.617 [*p* from 0.006 to 0.023].

Figure 6 and Figure 7 illustrate how results differ for mPE, HFD and conventional HRV HF and LF relative (%) power measures, for both ECG (RRi) and RSP (BBi) data. Results for PPG (PPi) data are not shown, as they were very similar to those for the RRi data.

The mPE measures show a similar decreasing patten for both RRi and BBi, but the Welch relative (%) HF and LF power measures shows opposite patterns for the two data types, with LF power increasing from normal breathing to breathing paced at 7 bpm for RRi, and HF power decreasing, whereas for BBi HF power increases and LF power decreases. Note that for the RRi data the slope of the line of HF power is less steep than that for the mPE measures, while for the BBi data this is the case for the line of LF power. Steeper slopes correspond to greater effect sizes.

The HFD measures are quite different for the RRi and BBi data, with less variation in slope for the latter. Note that for the RRi data the slope of the line of LF power is less steep than that for the HFD measures, while for the BBi data this is the case for the line of HF power.

#### 3.3.3. Conclusion of Paced Breathing Example Study

In conclusion, as anticipated, for all three data streams, complexity and entropy measures differentiated more clearly than conventional linear, time- and frequency-domain measures, between spontaneous and 7 bpm paced breathing. More measures decreased than increased, Higuchi fractal dimension at various lags being a notable exception.

## 4. Discussion

CEPS is a work in progress, and our dream is inevitably not yet fully materialised. We realise that there will be limitations to the use of CEPS which are dependent on the accuracy of the data used whether it be disease-related or simply based on normal physiological functioning. As a platform for analysis of Complexity and Entropy in Physiological Signals, CEPS meets a number of clinical research requirements for both large and small data samples. In future, other utilities may be added using a plug-in function (see Conclusions for future developments). As CEPS is issued under a creative commons license, it is free to use and can be modified and redistributed as long as the authors and this article are cited. This also applies to results that are generated with the GUI and are published in the future.

### 4.1. Limitations

As with all publications in a rapidly moving field the literature review that has informed the development of CEPS is continuously evolving, as is CEPS itself. It is worth stating that PubMed only provides a limited insight into patterns of publication but that a full literature review would have quite unmanageable in the time available to us and beyond the scope of this paper. As an illustration, whereas the number of papers in PubMed located using the search term ‘entropy’ was 34,647 in August 2020, before PubMed indexing of *Entropy* was rolled out in December 2020, this number had swelled to 39,571 by 23 January 2021. By that date the corresponding number for SCOPUS was 260,974, or more than 6.5 times greater, with around 3.5 million mentions in Google Scholar.

A particular technical limitation is that CEPS is not yet able to import data in all the various proprietary output formats by a burgeoning number of wearable technologies.

Certain functions are planned but remain to be fully implemented. In the Pre-processing mode, these include various methods of data interpolation, normalisation, coarse-graining, resampling and detrending. In the time-frequency domain, a number of spectral entropy methods have also not yet been implemented. Furthermore, while data segmentation is implemented, sliding window analysis is not yet a feature.

Inevitable technical glitches still exist—users are encouraged to report these. Users should also remember that CEPS is not magic, but a tool that must be used with care and discrimination.

### 4.2. Advantages

CEPS has been designed for the analysis of Complexity and Entropy in Physiological Signals intended to suit the requirements of clinical research on both large and small data samples. Although other such software packages exist (PyBioS and EZ Entropy are elegant examples), many are quite technical to use for non-specialist clinicians, and none include so many established or innovative complexity and entropy measures as CEPS. One particularly useful feature is the ability to test the effect of parameter variation on the different measures and to visualise how multiscaling may affect them. Another advantage is the facility for easy comparison of changes in different CEPS measures for the same or parallel physiological data streams, as we have done here in our biomedical engineering investigation of the effects of paced breathing on ECG, PPG and RSP variability.

## 5. Conclusions and Future Directions

We are aware that CEPS is not unique, and that some clinicians may find other packages more suited to their needs. In general terms, we have proven the utility of CEPS as a cross-platform (Linux, Mac or Windows based machines) MATLAB graphical user interface (GUI) which is more intuitive than command-line or menu-driven interfaces that rely on programming skills or are limited in the analysis options available. GUIs allow direct manipulation of graphical icons such as buttons, scroll bars, windows, tabs, menus, and cursors and allow the exchange of data between different software applications or data sets. They are much easier to use for beginners who do not have to learn command line routines, are easy to explore using the WIMP (windows, icons, menus, pointer) GUI interface provided by CEPS and provide a platform where the user can switch easily between tasks. Even using data with artefacts, we can tentatively conclude that CEPS has shown its usefulness. It has been possible, for instance, to process a welter of results to find a few measures that appear stable to ectopic beats or noise in RRi data. We are also pleased that use of CEPS has enabled us to confirm the general finding that complexity and entropy changes often show more significant changes than conventional linear measures (here, in the analysis of changes that are found during paced compared to spontaneous breathing). The general reduction in complexity in ECG and PPG variability during breathing at 7 bpm echoes that in RSP variability itself, but the latter also requires further investigation. The autonomic implications of the changes found also invite research. Future study, for which recruitment is currently underway, will explore the effects of paced breathing at different rates on a broader selection of CEPS measures and for a larger sample. We also plan to use CEPS in analysing the effects of different frequencies of peripheral electrical stimulation on EEG, HRV, respiration, postural sway and temperature data.

Two of our original objectives have not yet been met. Future developments of CEPS will include a ‘plug-in’ facility to allow other researchers to add measures not already included in the list available, and Classification will be added as a further item in the Application Mode drop-down list, providing alternative methods of classifying results from using CEPS or from other sources. AdaBoost, Sensitivity-Specificity and Linear discriminant analysis are being considered for inclusion in the next iteration of CEPS. No doubt other developments will be made in time and in response to user feedback.

Nonetheless, although still in process of development, CEPS is already a versatile tool for the analysis of complexity and entropy in physiological signals, and we feel justified in releasing it at this stage to invite feedback and foster awareness of the hidden mycelium that nourishes the burgeoning worlds of complexity and entropy.

## Figures and Tables

**Figure 1 entropy-23-00321-f001:**
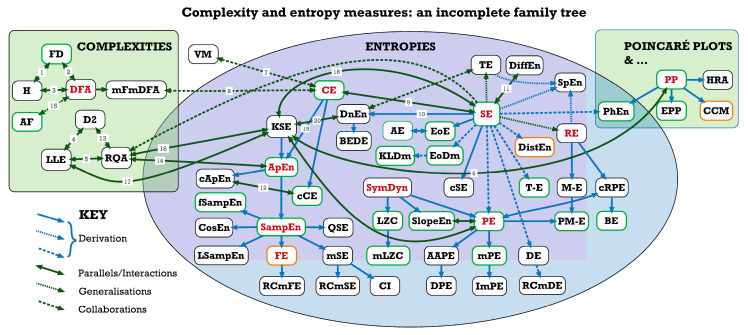
Complexity and entropy measures: an incomplete family tree. For abbreviations used, see the table at the end of the article. Green box outlines indicate results validated for CEPS, amber that they are acceptable but may not be 100% accurate, and black that results have not yet been validated for the implementations in CEPS (See Table 7).

**Figure 2 entropy-23-00321-f002:**
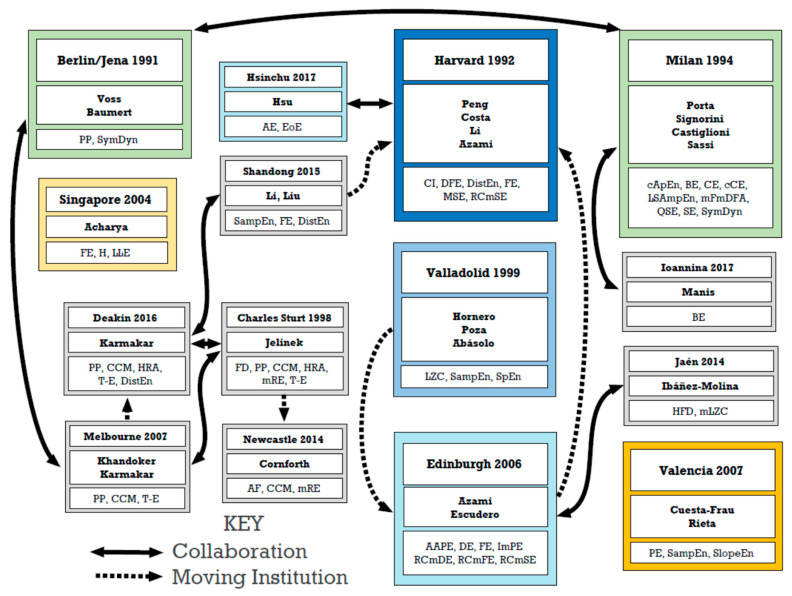
Hubs of learning—Major and linked hubs of learning for the complexity and entropy measures implemented in CEPS, as well as some others not included. Dates are of the earliest relevant studies by the named researchers. Arrows indicate collaborative publications and moves from one institution to another.

**Figure 3 entropy-23-00321-f003:**
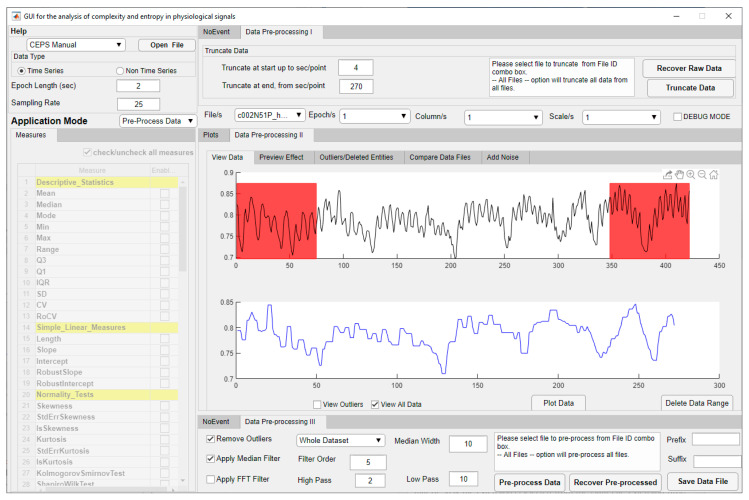
CEPS in Pre-Process Application Mode, showing raw PPG data above and the same data trimmed at either end (shown in red), filtered and with outliers removed below.

**Figure 4 entropy-23-00321-f004:**
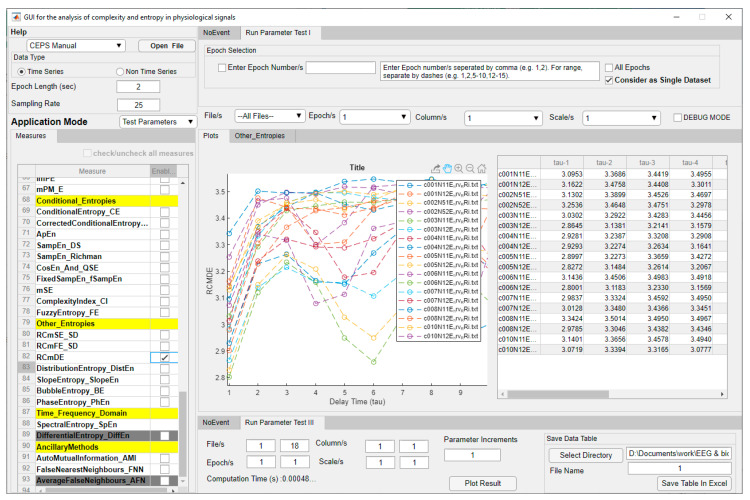
CEPS in Parameter Test Application Mode (‘Test and Plot’), showing an initial increase in RCmDE for ECG RRi data, followed by a decrease at around *τ* = 5 to 6 for some study participants but not others. Headings in the Measures list are shown in yellow. Measures greyed out are not yet implemented.

**Figure 5 entropy-23-00321-f005:**
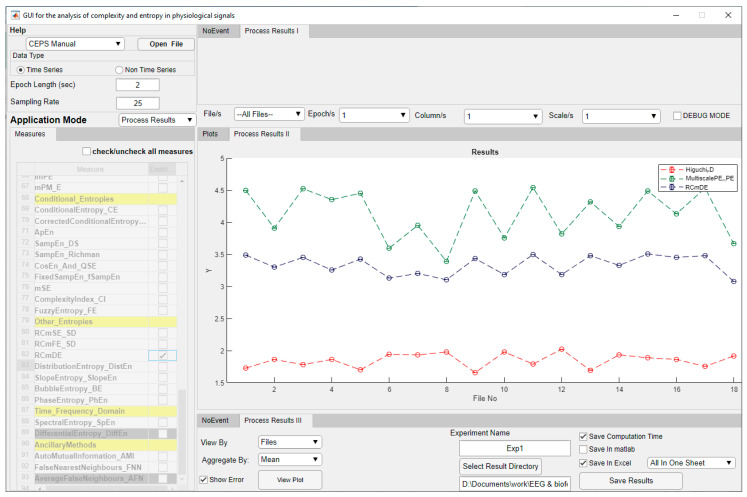
CEPS in Processing Results Application Mode, showing plots of Higuchi FD (*k_max_* = 13), mPE and RCmDE (both at lag 4) for ECG RRi data. For most study participants, HFD increases during paced breathing (i.e., at even file numbers), while mPE and RCmDE decrease (see too Figure 6). Note that the Measures list is greyed out when processing results.

**Figure 6 entropy-23-00321-f006:**
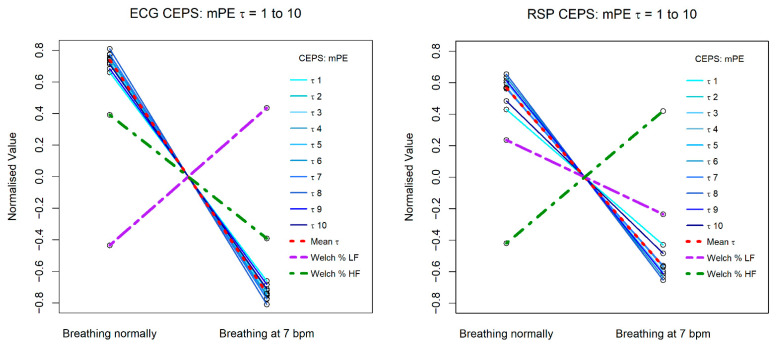
Change in CEPS mPE between normal breathing and paced breathing at 7 bpm, for values of lag τ from 1 to 10. The left plot shows results for ECG RRi data and the right for RSP BBi.

**Figure 7 entropy-23-00321-f007:**
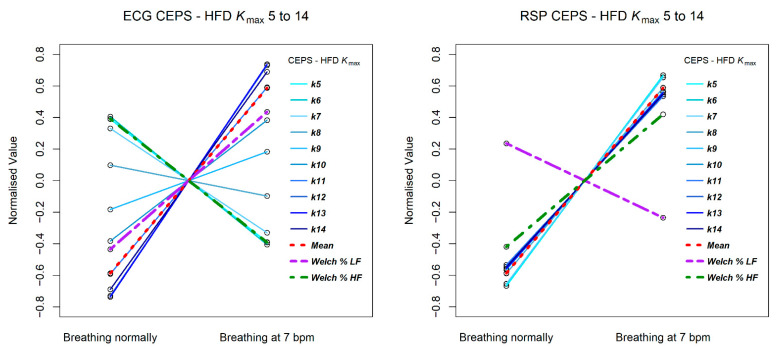
Change in CEPS HFD between normal breathing and paced breathing at 7 bpm, for values of k_max_ from 5 to 14. The left plot shows results for ECG RRi data and the right for RSP BBi.

**Table 1 entropy-23-00321-t001:** Major complexity and related measures sought in PubMed.

Measure	Abbreviated Name	Search Term
Higuchi fractal dimension	HFD	Higuchi AND “fractal dimension”
Katz fractal dimension	KFD	Katz AND “fractal dimension”
Allan Factor	AF	“Allan Factor”
Correlation dimension	D_2_	“Correlation dimension”
Hurst exponent	H	“Hurst exponent”
Detrended fluctuation analysis	DFA	“Detrended fluctuation analysis”
Largest Lyapunov exponent	LLE	Various ^1^
Recurrent quantification analysis	RQA	“Recurrent quantification analysis”
Poincaré plot	PP	“Poincaré plot” NOT “return plot”
Lempel-Ziv complexity	LZC	“Lempel-Ziv complexity”

^1^ “Largest Lyapunov Exponent” OR “maximal Lyapunov Exponent” OR “maximum Lyapunov Exponent” OR “greatest Lyapunov Exponent”.

**Table 2 entropy-23-00321-t002:** Major entropy measures sought in PubMed.

Measure	Abbreviated Name	SEARCH TERM
Shannon entropy	SE	“Shannon entropy”
Rényi entropy	RE	“Renyi entropy”
Min-entropy	M-E	“Min-entropy”
Tsallis entropy	TE	“Tsallis entropy”
Kolmogorov-Sinai entropy	KSE	“Kolmogorov entropy” or “Kolmogorov-Sinai entropy”
Permutation entropy	PE	“Permutation entropy”
Conditional entropy/ Corrected conditional entropy	CE/ CCE	“Conditional entropy”
Approximate entropy	ApEn	“Approximate entropy”
Sample entropy	SampEn	“Sample entropy”
Coefficient of Sample entropy	CosEn	“Coefficient of sample entropy”
Quadratic Sample entropy	QSE	“Quadratic sample entropy”
Multiscale entropy	MSE	“Multiscale entropy”
Fuzzy entropy	FE	“Fuzzy entropy”
Dispersion entropy	DE	“Dispersion entropy”
Slope entropy	SlopeEn	“Slope entropy”
Bubble entropy	BE	“Bubble entropy”
Distribution entropy	DistEn	“Distribution entropy”
Phase entropy	PhEn	“Phase entropy”
Spectral entropy	SpEn	“Spectral entropy”
Differential entropy	DiffEn	“Differential entropy”
Diffusion entropy	DnEn	“Diffusion entropy”
Symbolic Dynamics ^1^	SymDyn	“Symbolic Dynamics”

^1^ This method is not strictly speaking an Entropy measure but is included here for completeness.

**Table 3 entropy-23-00321-t003:** CEPS and Kubios HRV measures considered appropriate for 300-s ECG, PPG and RSP data. Numbers of multiple lags (*k*) used in this analysis are shown, as well as the number of measures in each category.

**CEPS (149 Measures)**		
**Category**	**Measures**	**Counts**
Descriptive measures	e.g., Mean, SD, CV, RoCV	12
Linear measures	e.g., Slope, Intercept, RoSlope, Skewness	7
Time domain	e.g., RMSSD, Hjorth A, M and C	5
Stationarity and c.	Auto-covariance (1-20), ACV	21
Complexity measures	HFD (5-14), H, RQA, EPP (1-10), CCM (1-10), LZC, mLZC, (1-9)	52
Shannon-based	SE, E-MC, AE, EoE, EoD*_m_*, KLD*_m_*, Tone	7
Ordinal entropies	mPE (1-10), ImPE (1-10), mPM-E (1-10)	30
Other entropies	RCmDE (1-10), DistEn, SlopeEn, BE, PhEn	14
**Kubios HRV (49 Measures)**		
**Category**	**Measures**	**Counts**
Time/Geometric domain	e.g., Mean HR, TINN, pNN50	11
Frequency domain	e.g., Welch and autoregressive peak Hz and band powers	34
Complexity	Poincaré SD1, SD2, SD1/SD2	3
Other	Stress index	1

For abbreviations, see table at the end of the Article.

**Table 4 entropy-23-00321-t004:** (A) Major complexity and related measures found in PubMed, with references for software packages known to include them; (B) Numbers of studies on measures found in PubMed on 6 August 2020 and in SCOPUS on 21 January 2021; (C) When each measure was first indexed in PubMed (reference in brackets); (D) Numbers of citations of that paper in Google Scholar (GS) and SCOPUS (S); (E) Peak year for number of occurrences (hits) in PubMed.

(A) Measure [Packages]	(B) *N* PubMed (SCOPUS) Hits	(C) Date of First PubMed Paper	(D) First Paper Citations—GS (S)	(E) Peak Year in PubMed
**HFD** [7,15,20,33,34,43]	118 (458)	1994 [44]	28 (17)	2019
**KFD** [15]	27 (135)	1994 [45]	43 (0)	2019
**AF** [22,33]	18 (98)	1996 [46]	152 (113)	2004–5
**D_2_** [5,20,22,23,29,30,31,33,43,47]	732 (3869)	1986 [48]	113 (70)	2019
**H** [5,15,20,33,43]	427 (3464)	1992 [49]	0 (2)	2008
**DFA** [10,15,20,21,22,23,24,26,27,33,34,43,47,50,51,52]	907 (3288)	1995 [53]	3850 (2619)	2017
**LLE** [5,8,15,22,23,29,30,33,43]	545 (2470)	1986 [54]	1081 (550)	2019
**RQA** [10,13,20,22,23,30,33,43,50]	386 (1201)	1997 [55]	154 (79)	2019
**PP** [7,10,13,15,20,21,23,27,30,33,43,47,50,56,57,58,59,60,61]	339 (870)	1992 [62]	472 (292)	2018
**LZC** [14,15,20,31,33,34]	201 (547)	1993 [63]	20 (16)	2015

**Table 5 entropy-23-00321-t005:** (A) Major entropy measures found in PubMed, with references for software packages known to include them; (B) Numbers of studies on measures found in PubMed on 6 August 2020 and in SCOPUS on 21 January 2021; (C) When each measure was first indexed in PubMed (reference in brackets); (D) Numbers of citations of that paper in Google Scholar (GS) and SCOPUS (S); (E) Peak year for number of occurrences (hits) in PubMed.

(A) Measure [Packages]	(B) *N* PubMed (SCOPUS) Hits	(C) Date of First PubMed Paper	(D) First Paper Citations—GS (S)	(E) Peak Year in PubMed
**SE** [8,15,34,64]	903 (6999)	1988 [65]	14 (11)	2019
**RE** [20,33,60]	138 (2704)	2000 [66]	64 (44)	2017/19
**M-E**	15 (500)	2012 [67]	172 (114)	2015–17
**TE** [34]	93 (1525)	2001 [68]	106 (89)	2015
**KSE**	129 (1121 ^a^)	1985 [69]	201 (119)	2000
**PE** [9,24]	234 (1347)	2002 [70]	2927 (1977)	2018
**CE/CCE** [9,15,20,33]	166 ^d^ (1635)	1998 [71]	286 (209)	2019
**ApEn** [9,13,15,20,22,23,24,28,31,33,43,62]	1199 ^b^ (2605)	1991 [72]	5394 (3801)	2013
**SampEn** [9,13,15,20,21,23,33,34,43]	1033 (2691)	2000 [73]	5712 (4126)	2018
**CoSEn**	8 (15)	2011 [74]	243 (183)	2018
**QSE**	4 (11)	2014 [75]	13 (11)	n/a (all tied)
**fSampEn**	9 (13)	2015 [76]	10 (6)	2018/20
**MSE** [13,24,28]	402 (959)	2002 [77]	2526 ^c^ (1843)	2018
**FE** [9,24]	121 (2003)	1998 [78]	118 (58)	2018
**DE** [24]	9 ^d^	2016 [79]	14 (11)	2017–20
**SlopeEn**	0 (3)	2019 [80]	1 (3)	n/a
**BE**	2 ^d^	2017 [81]	42 (34)	n/a (both tied)
**DistEn** [9,24]	31 ^d^	2015 [82]	129 (110)	2019
**PhEn** [24]	1 ^d^	2019 [83]	5 (3)	2019
**SpEn** [20,33,43]	302 (1022)	1991 [84]	26 (18)	2019
**DiffEn**	72 (942)	1970 [85]	44 (21)	2018
**DnEn**	31 (117)	2002 [86]	50 (27)	2016
**SymDyn** [15,20,24,33,34,87]	329 (2383)	1995 [88]	79 (55)	2015

**Notes:**^a^ Kolmogorov entropy (KE) = 696; KE *and* Kolmogorov-Sinai entropy = 1121; ^b^ On repeating this search 21.01.21 for the same period (i.e., to 6 August 2000), PubMed count had increased to 1234; ^c^ Two papers were published by Costa et al. in 2002—one in *Computers in Cardiology*, a relatively small circulation journal with no doi, one in *Physical Review Letters*, a more widely read journal with doi; ^d^ These are specifically citations of the Conditional entropy of Porta’s group, the Dispersion entropy created by Rostaghi and Azami, the Bubble entropy of Manis et al., the Distribution entropy of Li, the Phase entropy of Rohila and Sharma, and the SymDyn methods of Voss and Porta; other measures with those names are also mentioned in PubMed. ^e^ Note that since the original searches on 06.08.20, the journal *Entropy* has now been indexed in PubMed.

**Table 6 entropy-23-00321-t006:** Those researchers who have published more than 25 papers on complexity or entropy topics, listed by the number of different measures on which they have written and their Institutional affiliation when they first published (unchanged to date, unless noted otherwise in parentheses).

Author	*N* Measures	*N* Papers	Earliest	Institution
Maria Signorini	17	31	1994	Politecnico Milano
U Rajendra Acharya	16	53	2004	Ngee Ann Polytechnic
Roberto Hornero	14	102	1999	U Valladolid
Xiaoli Li	14	82 ^1^	2005	Beijing Normal U
Alberto Fernández	14	44	2006	U Complutense, Madrid
David Cuesta-Frau	14	28 ^2^	2007	U Politècnica, València
Alberto Porta	13	74	1994	U Brescia (now U Milano)
Andreas Voss	13	38	1991	Institut für Herz-Kreislauf-Forschung, Berlin (now U Applied Sciences, Jena)
Jamie Sleigh	13	37	1995	Waikato U (now Waikato Hospital)
Daniel Abásolo	12	57	2005	U Valladolid (now U Surrey)
Javier Escudero	12	52	2006	U Valladolid (now U Edinburgh)
Jesús Poza	8	35	2005	U Valladolid
Marimuthu Palaniswami	7	36	2001	U Melbourne
Ki Chon	7	34	2001	City U, NY (now U Connecticut)
Jiann-Shing Shieh	7	34	2009	Yuan Ze U (now National Taiwan U Hospital)
Nick Stergiou	6	85	2003	U Nebraska
Men-Tzung Lo	6	33	2010	National Taiwan U Hospital (now National Central U, Chungli)
Jack J Jiang	6	29	2001	U Wisconsin
Heikki Huikuri	5	49	1996	U Oulu
Timo Mäkikallio	5	30	1996	U Oulu
Chandan Karmakar	4	34	2007	U Melbourne (now Deakin U)
Steven Pincus	3	74	1991	Yale U (now Chapman U)
Metin Akay	3	35	1996	Rutgers U (now U Houston)
Chung-Kang Peng	2	49	1992	Boston U (now Harvard U)
Chengyu Liu	2	27	2011	Shandong U (now Southeast U)
Johannes Veldhuis	1	170	1994	U Virginia (now Mayo Clinic)
Ferdinand Roelfsema	1	60	1996	Leiden U
Ary Goldberger	1	46	1991	Beth Israel Hospital, Boston (now Harvard U)
Ali Iranmanesh	1	40	1996	Salem Veterans Affairs Medical Center
Marijke Frölich	1	28	1997	Leiden U
Yan Li	1	27 ^1^	2009?	U Southern Queensland

^1^ These numbers may not be accurate, as the family name Li is not uncommon, given names are not always included in publications, and many different given names start with the same letter. ^2^ This includes early publications under the name David Cuesta. U: University (or equivalent).

**Table 7 entropy-23-00321-t007:** Complexity and entropy measures in Figure 1 and their sources, listed in alphabetical order, showing original references, names of code providers, code type, institutions of originators and (in parentheses) of code providers, how code was verified, and whether verification showed optimal, acceptable or divergent results (O, A or D) according to the criteria used by Gomes [27], with optimal verification confirmed if Pearson’s *R* > 0.9; pD indicates partial divergence and T that testing is still required. Names asterisked (*) are of those who provided their own codes in response to personal requests (@ indicates no code provided in response).

Measure	Original Author/s	Provider	Source Code	Institution	Verification	O, A or D
AAPE	Azami and Escudero 2016 [109]	DataShare	MATLAB	Edinburgh	Cuesta-Frau/D.M.	D
AE	Hsu et al., 2019 [110]	Hsu *	MATLAB	Hsinchu	Hsu/D.M.	O
AF	Allan 1966 [111]	Cornforth *	C++	(Newcastle)	Cornforth/D.M.	O
ApEn	Pincus et al., 1991 [72]	Monge	matlabcentral	(Valladolid)	Rohila [HRVAnalysis, Kubios]	O [D]
BE	Manis et al., 2017 [82]	Manis *	python	Ioannina, Milan	Manis, Rohila, D.P.	O
BEDE	Qi and Yang 2011 [112]	@	n/a	Shanghai	n/a	
CE/CCE	Porta et al., 1998 [71]	Monge	matlabcentral	Milan (Valladolid)	[HRVAnalysis]	D/O
CCM	Karmakar et al., 2009 [113]	Cornforth *	C++	Melbourne (Newcastle)	Cornforth/D.M.	A
CI	Costa et al., 2008 [114]	D.P. (tbc)	MATLAB	(Hertfordshire)	Implementation in progress	T
CoSEn	Lake and Moorman 2011 [115]	D.P. (tbc)	MATLAB	(Hertfordshire)	Implementation in progress	T
D_2_	Theiler 1987 [116]	Faranda and Vaiente [117]	MATLAB	(Paris-Saclay, Aix Marseille)	Verification in progress	[D]
DE	Rostaghi and Azami 2016 [79]	DataShare	MATLAB	Edinburgh	PyBioS/tbc	T
DFA	Peng et al., 1994 [118]	Magris (tbc)	MATLAB	Harvard	Castiglioni/HRVAnalysis	T
DiffEn	Shi et al., 2013 [119]	D.P. (tbc)	MATLAB	(Hertfordshire)	Implementation in progress	T
DnEn	Grigolini et al., 2001 [120]	@	n/a	N Texas	n/a	
DistEn	Li et al., 2015 [82]	Li *	MATLAB	Harvard	[PyBioS]	[A]
DPE	Martínez-Rodrigo et al., 2019 [121]	@	n/a	Castilla-La Mancha (xxx)	n/a	
EoD*_m_*	Nardone 2014 [122]	Nardone *	mathematica	Bruxelles	Nardone/D.P.	O
EoE	Hsu et al., 2017 [123]	Hsu *	MATLAB	Hsinchu	Hsu/D.M.	O
EPP	Satti et al., 2019 [124]	Mani *	MATLAB	UCL	Mani [Kubios HRV, HRVAnalysis]	O [A]
FD	Higuchi 1988 [125]	Monge	matlabcentral	(Valladolid)	Ibáñez-Molina [HRVAnalysis]	A [O]
FE	Chen 2007 [126]	DataShare	MATLAB	Edinburgh	Rohila [PyBioS]	A [D]
fSampEn	Sarlabous et al., 2014 [127]	Estrada	MATLAB	Barcelona	Estrada and Torres	O
H	Hurst 1965 [128]	Davidson	matlabcentral	not known	[HRVAnalysis]	D
HRA	Jelinek et al., 2011 [129]	n/a	n/a	n/a	n/a	
ImPE	Azami and Escudero 2016 [130]	DataShare	MATLAB	Edinburgh	Verification in progress	T
KSE	Grassberger and Procaccia 1983 [131]	n/a	n/a	n/a	n/a	n/a
LLE	Rosenstein et al., 1993 [132]	Kizilkaya	matlabcentral	n/a	[HRVAnalysis]	[D]
LZC	Lempel and Ziv 1976 [133]	Thai	matlabcentral		Ibáñez-Molina	O (tbc)
mFmDFA	Castiglioni et al., 2019 [134]	Castiglioni *	MATLAB	Milan	Implementation in progress	T
mLZC	Ibáñez-Molina et al., 2015 [135]	Ibáñez-Molina *	MATLAB	Jaén	Ibáñez-Molina	O (tbc)
MSE	Costa et al., 2002 [77]		MATLAB	Harvard	Reinertsen/Da Poian/[PyBioS]	D
PE	Bandt and Pompe 2002 [70]	DataShare	MATLAB	(Edinburgh)	Rohila, Zunino [PyBioS]	O [D]
PhEn	Rohila and Sharma 2019 [83]	Rohila *	MATLAB	(Roorkee)	Rohila [PyBioS]	O [A]
PM-E	Zunino et al., 2015 [136]	Zunino *	MATLAB	La Plata	Zunino	O
QSE	Lake 2011 [74]	D.P. (tbc)	MATLAB	(Hertfordshire)	Implementation in progress	T
RE	Rényi 1961 [137]	Mathworks	matlabcentral	(Shanghai)	Verification in progress	T
RQA	Zbilut et al., 2002 [138]	Mathworks	matlabcentral		[Kubios HRV]	[D]
SampEn	Azami and Escudero 2016 [139]	DataShare	MATLAB	Edinburgh	Rohila [Kubios HRV, PyBioS]	O [pD ^1^]
SE	Shannon 1948 [97]	Mathworks	matlabcentral	various	[HRVAnalysis]	[O ^2^]
SlopeEn	Cuesta-Frau 2019 [80]	Cuesta-Frau *	MATLAB	Valencia	Cuesta-Frau/D.M.	O
SpEn	Inouye et al., 1991 [84]			tbc	Implementation in progress	T
SymDyn	Voss et al., 1995 [88]	tbc	tbc	tbc	Not yet implemented	
TE	Tsallis 1988 [140]	Guan	matlabcentral	(Shanghai)	Verification in progress	T
T-E	Oida et al., 1997 [141]	Karmakar	MATLAB	Kyoto	Karmakar/D.P.	O
VM	Bernaola-Galván et al., 2017 [40]	Bernaola-Galván *	Fortran, MATLAB	Málaga	Implementation in progress	T

^1^ Agreement was better for SampEn between CEPS (DataShare) and Kubios HRV than between CEPS (DataShare) and PyBioS or CEPS (DataShare) and HRVAnalysis, but results were partially divergent for all comparisons. However, agreement was better overall for SampEn between CEPS (DataShare) and Kubios HRV than between Kubios HRV and PyBioS or between Kubios HRV and HRVAnalysis. Agreement was good for various short samples of synthetic data (length 50 points) between results provided by Ashish Rohila and those from CEPS; ^2^ Results for a second implementation of SE (‘Entropy_MC’) are not identical and should be treated with caution.

**Table 8 entropy-23-00321-t008:** Results of a non-systematic literature review of the data requirements for measures in CEPS (further information and references available in the Primer accessed via the HELP section in CEPS). (A). Measure suited to discrete (d) data (e.g., RRi) and/or continuous (c) data (e.g., EEG); (B). Measure suited to short (s) (100 data points or even less), medium (m) (300–1000 points) and/or long (l) datasets (10,000 points or more); (C). Measure likely to be affected by noise (y or n); (D). Measure has been or could be used with bandpass (or other) filtered data (y or n), or likely to be affected by filtering (a); (E). Measure likely to be affected by sampling rate or down-sampling (y or n); (F). Whether measure suited to stationary/nonstationary (s/ns) or linear/nonlinear (l/nl) data. ‘T’ indicates that testing is required to clarify. Asterisked measures (*) are not implemented in CEPS.

Measure	A: d and/or c	B: s, m and/or l	C: Noise	D: Filtered	E: Sampling	F: s/ns, l/nl
AAPE	d, c ^1^	Ts, m, Tl	T	T	T	s, ns, l, nl
AE	d, Tc	s, m, Tl	y	T	y	Tns, nl
AF	d	l	T	T	T	s, Tnl
ApEn	d, c	l	y	y, a	y	s, nl
BE	d, c	s, m, l	n	T	T	Tns, nl
CE/CCE	d, c	Ts, m, Tl	y	a	y	s, nl
CCM	d	Ts, m, Tl	y	T	y	s, ns, l, nl
CI	d, c	l	y	T	y	s, nl
CosEn	d, c	s	y	a	y	s, nl
D_2_	d, c	l	y	y, T	y	s, nl
DE	Td, c	Ts, m, l	y	T	y	ns, nl
DFA	d, c	s, m, Tl	T	a	y	ns, nl
DiffEn	Td, c	m, l	T	y	T	s, Tns, l, Tnl
DnEn	d, c	l	T	T	T	s, nl
DistEn	d, c	s, m, Tl	T	T	y	ns, nl
EoD*_m_*	d, c	T	T	T	T	T
EoE	d, Tc	s, m, Tl	y	T	y	Tns, nl
EPP	d, c ^1^	s, m, l	T	y, a	y	s, ns, l, nl
FD	d, c	s, m, l	n	y, a	T	ns, nl
FE	d, c	s, m	n	T	T	ns, nl
fSampEn	d, c	m, Tl	T	y	y	s, nl
H	d, c	Ts, m, l	T	y, T	y	s, nl
IMPE	d, c ^1^	s, m, Tl	n	T	T	ns, nl
KSE *	Td, c	Ts, Tm, l	y	T	y	s, nl
LLE	d, c	m, l	y	y, a	y	s, nl
LSampEn *	d	s	n	T	T	s, nl
LZC	d, c	s, m, l	n	a	T	ns, nl
mFmDFA	d, c	s, m, l	T	y, a	T	ns, nl
mLZC	Td, c	s, m, Tl	T ^2^	T	T	ns, nl
MSE	d, c	m, Tl	y	y, a	y	s, nl
PE	d, c ^1^	s, m, l	n	y, T	n	ns, nl
PhEn	d, c	s, m, l	n ^3^	T	y	ns, nl
PM-E	d, c ^1^	s, m, l	n	T	T	ns, nl
QSE	d, c	s	y	y	y	s, nl
RCmDE	Td, c	s, m, l	n	y	y	ns, nl
RCmFE σ	Td, c	s, m, l	T	T	T	ns, nl
RCmSE σ	d, c	Tm, l	T	T	T	Tns, nl
RE	d	Ts, m, Tl	T	T	T	ns, nl
RQA	d, c	s, m, Tl	n	y, a	y	ns, nl
SampEn	d, c	m, Tl	y	y, a	y	s, nl
SE	d	s, m, l	y	T	T	s, nl
SlopeEn	d	s, m, l	n	y	T	ns, nl
SpEn	d, c	s, m, Tl	y	y	y	s, l
SymDyn	d, c	Ts, m, l	T	T	T	ns, nl
TE	d	Ts, Tm, Tl	T	T	T	Tns, nl
T-E	d	M ^4^	y	T	T	s, ns, nl
VM	d, Tc	s, m, l	T	T	T	ns, nl

^1^ Signal requires prior partitioning; this is implemented CEPS codes used for these variants of PE; ^2^ Effect of noise likely to depend on signal-to-noise ratio; ^3^ Effect of noise likely to be small; ^4^ In T-E, 250 data points are required for Tone estimation, but only 50 for Entropy [152].

## Data Availability

The data presented in this study will shortly be freely available in Open Research Data Online (ORDO), The Open University’s searchable research data repository at https://ordo.open.ac.uk/ (accessed on 5 February 2021).

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
