# Peer review of "CEPS: An Open Access MATLAB Graphical User Interface (GUI) for the Analysis of Complexity and Entropy in Physiological Signals"

_entropy, 2021, doi:10.3390/e23030321_

Round 1

Reviewer 1 Report

The first impression when I saw this manuscript was that it is way too long for a paper describing a program package for complexity and entropy analysis. The authors might be too ambitious, I would say, incorporating not only the introduction of the toolbox itself, but also the study involving heart rate variability data during and post COVID-19 infection. I strongly recommend the authors cut significantly the contents. A practical way to shorten it is probably to make the current paper two parts, with the first part describing their newly developed software application, and the second part for a study of heart rate variability during and post COVID-19. The reason that I suggest separating them is that the study of COVID-19 related heart rate variability changes is independent of any software packages. People can do the research without this package, and this package seems to be able to do a lot more than simply the study of heart rate variability. Definitely the authors can do a case description in order for readers to better understand the use of this toolbox—this does not need lengthy paragraphs to introduce background information, discussion, and interpretation of the results.

I only have a gut feeling that the authors might be too optimistic about their software package. I would recommend the authors root the introduction a bit more on the common questions, instead of being purely based on their own experiences. Different labs may have different styles to perform the tasks, and thus, what the authors’ lad requires may not necessarily reflects the needs of other labs. In other words, what the authors have tried to optimize here in this package may ultimately be not necessary for other labs because they have their own requirements and settings. For sure it definitely is appealing to talk about their very own stories and struggles in the past and how they finally ended up with this handy tool. But scientifically, I am awaiting to see any common interests that motivate the development of this package. Otherwise it looks that there are already quite a few software applications that have been developed and verified.

Some of the bullet points in objectives can be combined, and thus this section can be significantly shortened. For me, the goal is to establish a software application that perform entropy and complexity analysis of physiological time series. What the authors listed are more like the functions of this application.

The diagram that shows the named researchers related to entropy/complexity sounds interesting. The authors have done a good job reviewing the histories. Just a quick suggestion. The author Li under Harvard block (Peng Li I guess) moved from Shandong University, China, the institution where he developed the distribution entropy. This should be a town that is worthy of a separate block here since another research Liu (Chengyu Liu who appears also in Table 7) has also studied intensively entropy measures back then.

The order of Table 6 may need improvement to show how interest shifts, or a specific order that is easier to read. Currently it appears very random. The average hits per annum is not quite easy to follow. Considering recent interests among researchers, it might make sense to show an average citation number of the original paper proposing the specific algorithm in the past, say, five years. This is easily accessible through either Google Scholar or WoS.

Have the authors verified their implementation of the algorithms? A comparison of case results between their results and results obtained from other openly available toolboxes will help convince readers that the implementation is valid.

And lastly, the Discussion section appears to be a mix of software development side and COVID-19 related stuffs. Separation of the manuscript will help improve the logic throughout. Some of the claims in the advantage section are not quite appropriate. For example, many of the published applications with user interface are to facilitate the use for non-biomedical engineers that requires no knowledge of programming. The authors may want to tone down a bit.

Author Response

Please see the attachment ('CEPS resubmission cover letter.docx'), which is addressed to both reviewers.

Reviewer 2 Report

Summary of Paper:
=================

The authors present a new MATLAB toolbox (CEPS) for computing various complexity and entropy measures. This toolbox provides a GUI interface for pre-processing time series data, interactively setting tuning parameters for the provided measures, and outputting the measures for later analysis. The authors then demonstrate CEPS on a data set consisting of Pulse Rate Variability (PRV) recordings for 13 individuals who self-reported as having COVID-19 and had PRV measurements before, during, and after the self-diagnosis. They find various differences in the complexity and entropy measures within-individuals between different phases of the COVID-19 illness process.

Overall Review:
===============

This is a very ambitious paper, and the CEPS toolbox seems like a perfect tool for non-technical researchers who would like to explore complexity and entropy measures with their data sets in the absence of scripting and / or programming expertise.

My main concern with the paper is that it is too ambitious. The paper combines a literature review on complexity and entropy measures, a mini-manual on CEPS, a literature review on associations of HRV with immune function, and the actual analysis done by the authors of the Welltory data set. This could be at least 3 publications, rather than a single publication. I feel that the paper is trying to do too much, and in the process does not do a satisfactory job at each part. For example, the complexity and entropy measures are listed and their interrelationships are given, but no attempt it made to discriminate between which measures might be best, and the end-user of CEPS is left to just "try everything," which could lead to hypothesizing after the results are known (HARKing), P-hacking, etc., without further guidance on how to focus their analysis. Similarly with the statistical analysis of the Welltory data: no attempt is made to address confounding of changes in complexity measures with time of day, the number of measurements taken, the device used, etc.

If the authors do decide to break up this paper into a series of more manageable papers, I also have the following comments and suggestions.

Comments and Suggestions:
=========================

1. With only 13 individuals considered, a convenience sample of those who have data before and after self-diagnosis who are all self-reporting COVID-19 using in-the-wild PRV measurement from a smart phone, I'm not sure how much insight can be gleaned from the Welltory data. For example, the fact that 701 individuals self-reported as having COVID-19 but only 13 of the 701 had PRV recordings before and after self-diagnosis indicates these 13 individuals are likely different from the 688 other individuals in the database, let alone a typical person diagnosed with COVID-19.

2. At several points, you indicate that differences between the measures at different phases of illness were "significant". Do you mean statistically significant? And if so, what is your significance threshold?

3. How is eta calculated? Do you mean the square root of the standard eta-squared statistic, e.g.

https://en.wikipedia.org/wiki/Effect_size#Eta-squared_(η2)

or something else? An easier to interpret (and perhaps more standard) effect size to use for differences computed using a Mann-Whitney (Wilcoxon rank sum) test would be the Hodges–Lehmann estimator.

3. The Kruskal-Wallis and Mann-Whitney tests assume that observations in each group are independent. That may not be the case for your data, where for example you have up to 193 observations in a single condition for certain participants. Are these observations in any way correlated, i.e. possibly all taken in a short period of time? This will invalidate the assumptions of the KW and MW tests.

4. The data would be better analyzed via a multilevel (e.g. mixed effects) model, rather than computing effect sizes for each individual and then aggregating them via a median or IQR. For example, one individual has only 6 PRV sessions, while another has 405. The effect sizes computed for these two individuals are hardly comparable: the first is much noisier than the second, and should be weighted accordingly. A multilevel model would do this explicitly.

5. With results reported for 26 measures in Table 11, it is important to make corrections for multiple comparisons. Many of these "large" effect sizes may be due to the fact that so many measures are being considered, rather than actual large effects in the population. It would thus be more useful to report confidence intervals for the effect sizes, adjusted for 26 comparisons, or use a method such as the False Discovery Rate.

Minor Comments:
===============

1. On line 334, you say that you "[test] data for normality of distribution." Here you presumably mean the "data" to be the complexity and entropy measures, and not the PRV dataset. Also, normality tests are not for the normality of **data**. As with all inferential tests, they are testing a property of the population from which the data arose. And you did not report on the results of the normality tests? Presumably, since you used nonparametric tests, you concluded the population(s) were non-normal? And did you do this testing per-individual, across individual, etc.? And if the latter, how did you account for the nested nature of the data?

Author Response

(The authors gave the same response as above.)

Round 2

Reviewer 1 Report

The authors have done a good job addressing my previous concerns. I'd like to endorse the publication, although I still have a strong feeling that the manuscript can still be further shortened for better readability. But I have to agree with the authors that Entropy, as a typical OA journal, has no page limit that allows authors to provide many details.

Author Response

Please see attached file, CEPS resubmission cover letter 2.docx 

Reviewer 2 Report

Overall Review:

The revisions to your paper in terms of scope and content have improved the paper's readability and in my opinion made it more likely to have a positive impact on the research of clinicians (and others) who would like to use entropy and complexity measures in their own work.

The statistical analysis has been clarified and strengthened by the inclusion of the explicit definition of their significance level and the use of the Benjamini–Hochberg procedure for controlling their False Discovery Rate. I do, however, have a few minor questions about the statistical analysis used, the clarification of which I believe will further improve the paper.

After the authors address these comments, I believe the paper is ready for publication.

Comments and Suggestions:

Line 273: "Bootstrapped t-tests were used to check..."

I assume you are bootstrapping for a paired t-test? There is no one "t-test," so please indicate which t-test you used. Your Figures 6 and 8 make it seem like you did a two- (independent) samples t-test, which would be inappropriate for your repeated measures data.

Similarly, there is no one way to bootstrap. Did you resample the difference scores? Did you use a case-resampling bootstrap, a parametric bootstrap, something else, etc.? Please indicate what type of bootstrap you used.

How many bootstrap samples did you use?

Once you have your bootstrapped test statistics, there are many ways to compute a P-value. Did you use a basic bootstrap P-value, a percentile-based P-value, a Normal theory P-value, a BCa P-value, etc.? Please clarify this point.

Line 645: "Significantly more measures decreased than increased [...] using the Binomial test."

The binomial test assumes that the "trials" (in your case, whether a given measure showed positive or negative change) are independent. This is almost certainly not the case for your data, since many of the complexity and entropy measures will be positively correlated. A binomial test is therefore not appropriate for your data. I recommend removing this sentence entirely, since the analysis is inappropriate and removing it does not change the main results of your paper.

Author Response

Please see attached file, CEPS resubmission cover letter 2.docx.
